# Experimental Study on the Effect of Root Content on the Shear Strength of Root–Soil Composite with Thick and Fine Roots of *Cryptomeria japonica* (Thunb. ex L.f.) D.Don

Jianping Liu [1,*], Yusha Tang [1], Yulin Jiang [1], Shixin Luo [1], Kai Wu [2], Xingxin Peng [3] and Yucong Pan [4]

1. College of Water Conservancy and Hydropower Engineering, Sichuan Agricultural University, Ya'an 625014, China
2. Sichuan Highway Planning, Survey, Design and Research Institute Ltd., Chengdu 610041, China
3. China Railway 11th Bureau Group Fourth Engineering Co., Ltd., Wuhan 430074, China
4. School of Civil Engineering, Wuhan University, Wuhan 430072, China
* Correspondence: jpliu_whrsm@foxmail.com

**Abstract:** The current research on slope protection with plants mainly focuses on herbs and shrubs. In order to investigate the difference in shear strength of root–soil composite with thick and fine roots under different root content conditions, *Cryptomeria japonica* (Thunb. ex L.f.) D.Don was selected as the research object, and the distribution characteristics of its roots with the increase of buried depth were studied using the longitudinal profile method. Based on the distribution range of root area ratio (RAR) in field investigation, the modified large-scale direct shear tests were executed on the root–soil composite samples with thick and fine roots of five *RAR* grades under four normal stress levels, and the variation rule of the shear strength of the root–soil composites with thick and fine roots under different root contents was analyzed. The influence mechanism of RAR was briefly discussed. The results show that the *RAR* of *Cryptomeria japonica* (Thunb. ex L.f.) D.Don increases first and then decreases with the increase of buried depth, and decreases with the increase of horizontal distance from the excavation point to the trunk. Both the thick and fine roots can increase the soil shear strength, but the effect of thick roots is greater than that of fine roots. The shear strengths of root–soil composites with thick and fine roots both increase first and then decrease with the increase of *RAR*, which means that there exists the optimal root content for the roots with the best reinforcement effect on soil, and the optimal *RAR* is 0.1% and 0.2%, respectively. The cohesive and internal friction angle of the root–soil composite can be improved by the roots of *Cryptomeria japonica* (Thunb. ex L.f.) D.Don, and the thick root is better than the fine root. Meanwhile, the enhancement effect on the cohesion is greater than that on the internal friction angle. The results are of great significance for understanding the effect of roots on soil shear strength and enriching the existing theory of slope protection with arbors.

**Keywords:** slope engineering; *Cryptomeria japonica* (Thunb. ex L.f.) D.Don; root area ratio; root–soil composite; shear strength

## 1. Introduction

With the continuous advancement of infrastructure development in China, the increasing intensity and frequency of extreme weather events have led to frequent occurrence of geologic disasters such as mudslides, collapses, and landslides [1–5]. The use of plants to maintain slope stability and prevent soil erosion has been recognized as an effective and sustainable measure and has been widely used [6–9]. Plant reinforcement of slopes mainly treats root systems and soil as a composite material, and when the root–soil composite is subjected to shear stress, the root system will mobilize its tensile strength and transfer the shear stress of soil to the root system, thus improving the soil shear strength [10]. The shear strength of the root–soil composite is one of the important indexes to study the

mechanical effect of soil consolidation and slope protection with root systems, and mainly it is determined by the tensile strength of the roots, the strength of the soil, and the friction strength of the root–soil interface together [11]. When the root diameter is small, the tensile strength of the root itself is lower than the friction strength of the root–soil interface, which is manifested as the pulling off of the root; when the root diameter is large, the tensile strength of the root itself is higher than the friction strength of the root–soil interface, which is manifested as the pulling out of the root. It can be seen that the thick roots and fine roots of plants show different ways of improving the soil shear strength, and the effect is not the same.

A large number of studies on the mechanical effect of soil consolidation and slope protection with root systems have been conducted by researchers at present, and they mainly focus on the tensile properties of roots and the shear strength of root–soil composite. Existing studies have certified that the tensile capacity of plant roots has a key influence on the root–soil interaction, and the stronger the tensile capacity of plant roots, the better the reinforcement effect of roots on slopes. Therefore, based on the root pulling test, a series of studies have been conducted on the tensile properties of roots. Some scholars have studied the relationship between the root diameter and the maximum tensile force of different kinds of plants (e.g., *Rosa canina* L., *Dittrichia viscosa* (L.) Greuter, *Spartium junceum* L., *Cotoneaster dammeri* C.K.Schneid., *Juniperus horizontalis* Moench, *Cynodon dactylon* (L.) Pers., *Picea asperata Mast.*, *Broussonetia papyrifera* (L.) L'Hér. ex Vent., *Bothriochloa ischaemum* (L.) Keng. *(B. ischaemum)*, *Carex tristachya* Thunb. *(C. tristachya)*, *Artemisia gmelinii* Web. *A. gmelinii)*, *Artemisia giraldii* Pamp. *(A. giraldii)*), and results show that the larger the root diameter, the greater the maximum tensile force [12–14], and the relationship between the two is a power function [13,15]. Some scholars have investigated the relationship between root diameter and tensile strength, and results show that the tensile strength decreased as a power function with the increase of root diameter [14,16–18]. In addition, Mahannopkul and Jotisankasa [19] found that the tensile strength of *Vetiveria zizanioides* (L.) Nash is affected by root water content and soil suction. Giadrossich et al. [20] found that the root arrangement significantly affects the maximum tensile strength of root–soil composite of *Picea abies* (L.) H.Karst., and the parallel arrangement is greater than the cross arrangement.

Slope instability is mainly caused by shear failure; however, the shear resistance of soil itself is limited. Plant roots penetrate the soil, bind closely to the soil particles, absorb their water, and form a fine root web within it, forming a root–soil composite, which is more resistant to shear stress [21–23]. Researchers mainly study the influence of different factors on the shear strength of root–soil composite through direct shear test and triaxial test at present. Some scholars have studied the effects of different kinds of plants (e.g., *Pinus massoniana* Lamb., *Rosa laevigata* Michx., *Smilax china* L., *Dicranopteris linearis* (Burm.f.) Underw., *Pyracantha fortuneana* (Maxim.) H.L.Li, *Robinia pseudoacacia* L., *Zea mays* Linn., *Solanum tuberosum* L., *Setaria italica* (L.) Beauv. et al.) on the shear strength of root–soil composite and found that the increase of the shear strength of root–soil composite is mainly due to the increase of cohesion, but the internal friction angle does not change much [24,25]. Some scholars have investigated the relationship between water content and shear strength of root–soil composite and found that with the increase of water content, the cohesion decreases and thus the shear strength decreases also [24,26,27]. In addition, Hu et al. [28] investigated the relationship between root diameter and shear strength of root–soil composite of five different plants (*Atriplex canescens* (Pursh) Nutt., *Caragana korshinskii* Kom., *Zygophyllum xanthoxylon* (Bunge) Maxim., *Nitraria sibirica* (DC.) Pall., *Lycium chinense* Mill.) and found that the shear strength of root–soil composite increases as a power function with the increase of root diameter. Bourrier et al. [29] established a numerical model of direct shear test of root–soil composite based on the discrete element method and found that the shear strength of root–soil composite increases with the increase of root content. Xu et al. [30] conducted a direct shear test simulation of root–soil composite of *Lolium perenne* L. with different buried depth of soil based on FLAC 3D and found that its shear strength increases first and then decreases with the increase of buried depth.

Zhu et al. [31] conducted a direct shear test on the root–soil composite of *Oryza sativa* L. with different root content and found that the cohesion and internal friction angle of the root–soil composite are both minimized when the root content is 1.1%. Valizade and Tabarsa [32] conducted a direct shear test of undisturbed soil and investigated the effect of different root contents on the shear strength of root–soil composite of *Vetiveria zizanioides* (L.) Nash and found that the optimum root content is 3% to 5%. Zhu et al. [33] conducted a large-box direct shear test on young plants of six species and compared their shear strength with that calculated by the Wu and Waldron model and found that the calculated value is 82.00%~98.39% higher than the measured value.

The above literature is summarized in Table 1.

**Table 1.** Statistical table of root–soil composite for soil consolidation and slope protection.

| Type of Test | Year | Source | Plants | Plant Classification |
|---|---|---|---|---|
| Root pulling test | 2007 | [15] | *Betula nigra* | Arbor |
| | 2007 | [16] | *Rosa canina* L., *Dittrichia viscosa* (L.) Greuter, *Spartium junceum* L. | Shrub |
| | 2010 | [17] | *Rosa canina* L., *Cotoneaster dammeri* C.K.Schneid., *Juniperus horizontalis* Moench | Shrub |
| | 2013 | [20] | *Picea abies* (L.) H.Karst. | Arbor |
| | 2016 | [13] | *Picea asperata* Mast. | Arbor |
| | 2019 | [18] | *Cynodon dactylon* (L.) Pers. | Herb |
| | 2019 | [19] | *Vetiveria zizanioides* (L.) Nash | Herb |
| | 2021 | [14] | *Broussonetia papyrifera* (L.) L'Hér. ex Vent. | Shrub |
| | 2022 | [12] | *Bothriochloa ischaemum* (L.) Keng. (*B. ischaemum*), *Carex tristachya* Thunb. (*C. tristachya*), *Artemisia gmelinii* Web. (*A. gmelinii*), and *Artemisia giraldii* Pamp. (*A. giraldii*) | Herb |
| Direct shear test or triaxial test | 2010 | [24] | *Robinia pseucdoacaia* L. | Arbor |
| | 2013 | [28] | *Atriplex canescens* (Pursh) Nutt., *Caragana korshinskii* Kom., *Zygophyllum xanthoxylon* (Bunge) Maxim., *Nitraria sibirica* (DC.) Pall., *Lycium chinense* Mill. | Herb |
| | 2014 | [33] | *Pinus massoniana* Lamb. et al. | Arbor |
| | 2021 | [30] | *Lolium perenne* L. | Herb |
| | 2022 | [25] | *Pyracantha fortuneana* (Maxim.) H.L.Li | Herb |
| | 2022 | [31] | *Oryza sativa* L. | Herb |
| | 2022 | [32] | *Vetiveria zizanioides* (L.) Nash | Herb |
| | 2023 | [23] | *Festuca arundinacea*, *Bromus inermis*, *Bromus tomentellus* | Herb |
| | 2023 | [26] | *Zea mays* Linn., *Solanum tuberosum* L., *Setaria italica* (L.) Beauv. et al. | Herb |
| | 2024 | [21] | *Agrostis stolonifera*, *Cynodon dactylon*, *Festuca arundinacea* | Herb |
| | 2024 | [22] | *Chrysopogon zizanioides* L. | Herb |

In summary, the current research on slope protection with plants mainly focuses on herbs and shrubs, and there are still relatively few studies on root diameter and shear strength of arbors with different root contents. Herbs and shrubs have short root systems and limited slope reinforcement effect, while the slopes with arbors are more stable and less sensitive to climate and soil [34], so it is particularly important to carry out theoretical research on slope protection with arbors. In addition, there is no differential study on the effect of different root contents on the shear strength of root–soil composites of arbors with thick and fine roots. Therefore, *Cryptomeria japonica* (Thunb. ex L.f.) D.Don, a typical arbor, was selected as the research object in this paper. The distribution characteristics of roots with the increase of buried depth were investigated by longitudinal profile method. By using the measured *RAR* to characterize the root content, the effects of different root contents on the shear strength of root–soil composites of *Cryptomeria japonica* (Thunb. ex L.f.) D.Don with thick and fine roots were investigated through improved large-box direct shear tests, and the variation rules of cohesion and internal friction angle of root–soil composites with thick and fine roots were analyzed. Finally, the influence mechanism of RAR was briefly discussed. The results are of great significance for understanding the

influence of roots on soil shear strength and enriching the existing theory of slope protection with arbors.

## 2. Overview of the Test Area

The test area is located in the Boshan Mountain, Ya'an City, Sichuan Province, as shown in Figure 1, which is a low hill at the junction of Chengdu Plain and West Sichuan Plateau. Its geographical coordinates are between the north latitude of 29°58′30″~29°58′42″ and east longitude of 102°59′30″~102°59′48″, with a total length of about 1300 m from east to west, a maximum width of 430 m from north to south, and an altitude of 600~750 m. It is a subtropical humid monsoon climatic zone, with rainfall concentrated in summer and at night. The annual rainfall is more than 2000 mm, the annual average evaporation is 838.8 mm, and the soil is purple soil. At the end of the 1970s, the test area began to implement soil and water conservation measures such as mountain closure and forest cultivation, planting, and protection. At present, the vegetation coverage rate of the hillside is more than 95%, and more than 50 kinds of mixed forest are cultivated. *Cryptomeria japonica* (Thunb. ex L.f.) D.Don, an evergreen tree with fast-growing and well-developed roots, was selected as the research object in the test area. It is a common afforestation and slope protection species in Sichuan Basin. The overall slope gradient of the test area is about 30°, with a southwesterly orientation.

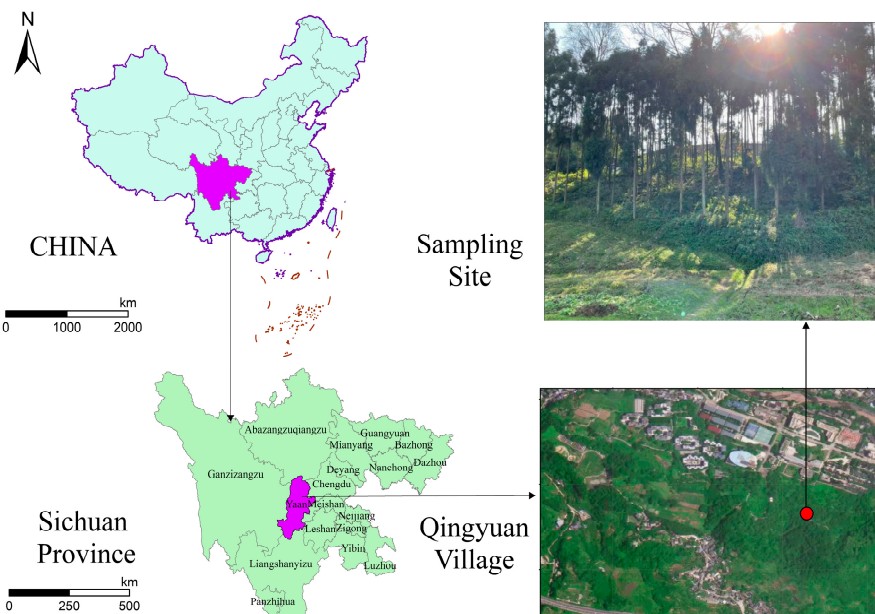

**Figure 1.** Position of Boshan Mountain.

## 3. Overview of the Test

### 3.1. Measurement Statistics of Root Content

Since the tree age could not be accurately estimated, three *Cryptomeria japonica* (Thunb. ex L.f.) D.Don with similar surface growth and no other trees around were selected as the research objects on the same slope. Their average diameter at breast height (DBH) is 16 cm, average plant height is 13 m, and average crown breadth is 2 m. The longitudinal profile method [35] was used to measure the root distribution on two longitudinal profiles with a horizontal distance of 0.5 m and 1 m from the tree trunk, respectively, as shown in Figure 2. The specific measurement method is as follows: with the tree trunk as the central point, an excavation belt with a width of 1 m is selected along the slope at 0.5 m and 1 m away from the central point, and the excavation is carried out vertically downward. In order to avoid damage to roots during excavation, manual excavation was adopted, and the excavation was carried out step by step every 20 cm until there was no root on the excavated profile. The overall excavation depth is about 1 m. Each excavation stage was calibrated from left

to right with a 20 cm × 20 cm square frame, and the diameter of root system was measured with a vernier caliper in the square frame.

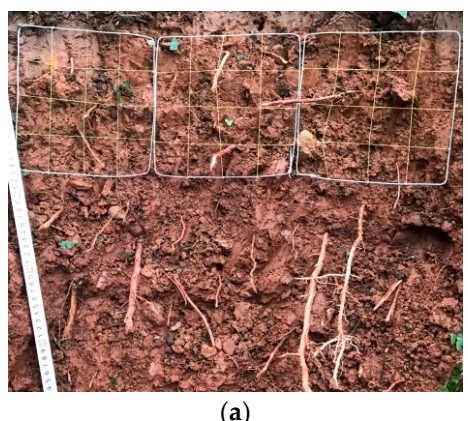 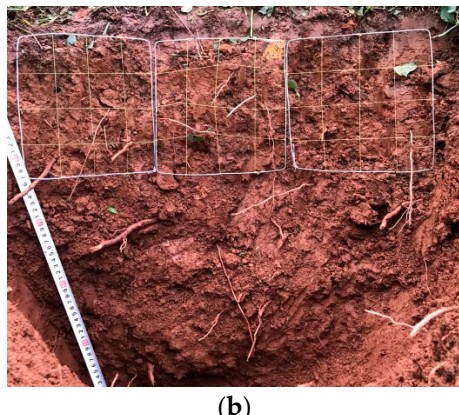

| (**a**) | (**b**) |

**Figure 2.** Diagram of longitudinal profile method: (**a**) Profile 1 and (**b**) Profile 2.

*3.2. Specimen Productio*

3.2.1. Soil Sample Material

The soil used in the test was taken from the surface soil in the root investigation area of *Cryptomeria japonica* (Thunb. ex L.f.) D.Don. The purple soil in the test area has the texture characteristics of loam and sandy loam, which is mainly agglomeration structure, medium hardness, good water drainage, ventilation, and nutrient retention ability, and can provide a suitable growth environment for plants. The basic physical properties of the soil at the sampling site were determined in accordance with the Standard for Geotechnical Testing Method (GB/T 50123-2019) [36]. Table 2 presents the basic physical properties of the soil.

**Table 2.** Basic physical properties of soil at the sampling site.

| Soil Type | Water Content $\omega$ /% | Wet Density $\rho$ /(g/cm³) | Dry Density $\rho_d$ /(g/cm³) | Liquid Limit $\omega_L$ /% | Plastic Limit $\omega_P$ /% |
|---|---|---|---|---|---|
| clay | 20 | 1.6 | 1.46 | 40 | 22 |

3.2.2. Root Material

Fresh roots of *Cryptomeria japonica* (Thunb. ex L.f.) D.Don with no surface damage were selected at the sampling site and washed in the laboratory as root materials for later tests. The internal size of the sample box provided for the test was 15 cm × 15 cm × 10 cm. Roots with uniform diameter distribution were selected and cut into 9 cm sections. In this way, the upper and lower ends of the roots were ensured to be 0.5 cm away from the upper and lower surfaces of the root–soil composite, and it can prevent the vertical deformation of the sample caused by the application of normal stress from pressing the roots and causing root bending, which can affect the cementation of the root–soil contact surface. In order to prevent the roots from drying out and affecting the test results, the roots were wrapped in plastic wrap and wet towel and stored in a dark place.

3.2.3. Sample Making Process

The root–soil composite samples were made according to the compression sample method of the remodeled soil sample preparation method in the Standard for Geotechnical Testing Method (GB/T 50123-2019) [36]. Using the principle of dry density control, two samples were made at a time by using a self-designed four-sided detachable sample box, and the size of a single sample was 15 cm × 15 cm × 10 cm. The roots were placed perpendicular to the shear plane and mutually parallel. The specific operation process is as follows.

The soil blocks retrieved from the investigation area were cut up and dried at 105 °C for more than 24 h to a constant weight, and then crushed with a wood roller and passed through a 2 mm sieve for later use. Use the dry density and natural water content of the soil at the sampling site to calculate the dry soil mass and the amount of water required for the remodeled soil sample. The corresponding mass of dry soil was weighed and placed on a plate. Use a pressure watering can to fill the required water, and slowly spray into the soil-filled plate, and repeatedly stir evenly. Wrap the plate with plastic wrap and leave it for one day and night to make the water content of the sample uniform. The roots and soil were filled into the sample box in layers, as shown in Figure 3a. The sample box was placed under a press to reach the required dry density of the sample, and left for 0.5 h to prevent the soil from rebounding, as shown in Figure 3b. Remove the sample box, disassemble the four sides of the box, cut the sample in half along the center line of the sample, and immediately wrap it with plastic wrap for later use, as shown in Figure 3c.

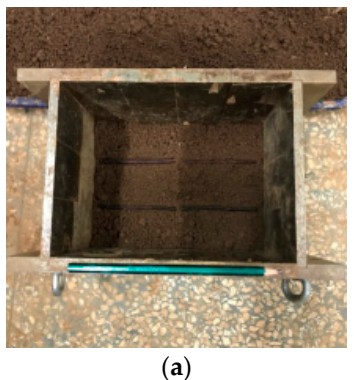 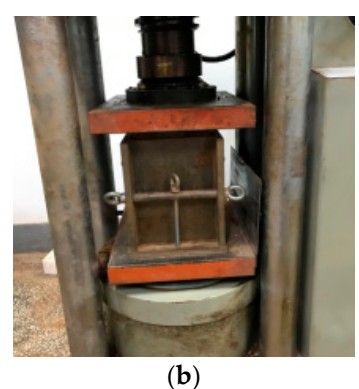 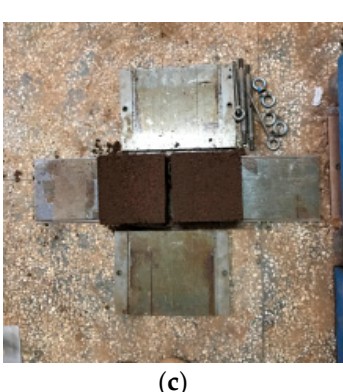

(**a**) (**b**) (**c**)

**Figure 3.** Sample preparation process of root–soil composite: (**a**) roots arrangement; (**b**) compression sample; and (**c**) cutting sample.

### 3.3. Test Equipment

As the sample used in the conventional direct shear test instrument for soil is too small to meet the requirements, based on the XJ-1 direct shear instrument for rock (Beijing China Coal Mine Engineering Co. Ltd., Beijing, China), an improved large-box direct shear test instrument was made in this study through redesigning the sample preparation device, replacing the digital pressure gauge with high precision but low range ones. The test instrument is mainly composed of three parts: upper and lower shear box, loading system, and displacement measurement system, as shown in Figure 4. As seen in Figure 4, the loading system is composed of hydraulic jacks, hydraulic oil pumps, and digital pressure gauges, and the displacement measurement system consists of dial indicators with an accuracy of 0.01 mm and magnetic bases. The sample deforms along the vertical direction after the normal stress is applied, which makes the upper and lower shear box easy to touch, resulting in test failure. In order to reduce the vertical deformation of the sample and better control the height of the gap between the upper and lower shear boxes, the height of the sample was reduced from the original 20 cm to 10 cm. At the same time, according to the vertical deformation of the sample under different normal stresses, wood blocks with a cross-section size of 15 cm × 15 cm and different heights were inserted into the upper and lower shear boxes, respectively. The height of the gap between the upper and lower shear boxes is always controlled at about 0.5 cm during the loading process.

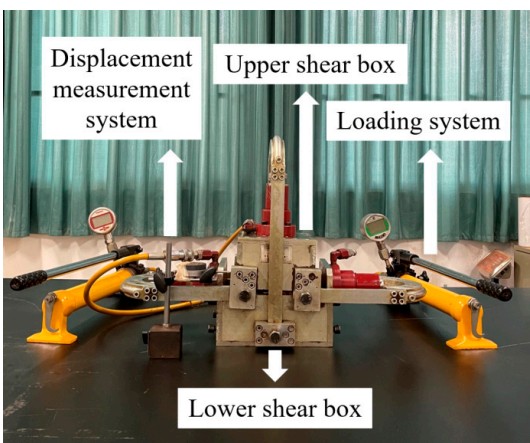

**Figure 4.** Equipment of large-box direct shear test.

*3.4. Test Program*

Root area ratio (RAR) is currently the main parameter for describing the root content in soil and is calculated using the following formula [35]:

$$RAR = \frac{A_r}{A} = \frac{\sum\limits_{i=1}^{n} \pi d_i^2 \bigg/ 4}{A} \tag{1}$$

where $A_r$ is the cross-sectional area of the roots (cm$^2$), $A$ is the sample area (cm$^2$), $d_i$ is the root diameter (mm), and $n$ is the number of roots.

There is no uniform standard for the grading of root diameter, and 2 mm or 5 mm is usually used as the boundary between thick and fine roots. Combined with the actual investigation results of root diameter of *Cryptomeria japonica* (Thunb. ex L.f.) D.Don, 5 mm was chosen as the grading limit of root diameter of thick and fine roots in this study [37]. The number and diameter of roots were used to calculate the *RAR*. According to the *RAR* range obtained from field measurements, five levels of *RAR* with a 0.1% gradient were set for the root–soil composites with thick and fine roots, as shown in Table 3. Large-box direct shear tests were carried out, respectively, under four normal stresses of 50, 100, 150, and 200 kPa for each *RAR* level.

**Table 3.** Gradient of *RAR*.

| Gradient Type | *RAR*/% | Root Number | | Root Diameter Range/mm | |
|:---:|:---:|:---:|:---:|:---:|:---:|
| | | Thick Root | Fine Root | Thick Root | Fine Root |
| I | 0.0 | | | | |
| II | 0.1 | 1 | 4 | >5 | <5 |
| III | 0.2 | 2 | 6 | >5 | <5 |
| IV | 0.3 | 3 | 8 | >5 | <5 |
| V | 0.4 | 4 | 12 | >5 | <5 |

Note: Different *RAR* are the results of individual calculations for thick or fine roots.

The large-box direct shear test was carried out by fast shear method. The shear stress was applied by stress control method, and the normal stress was kept constant during the test. In order to prevent sudden shear failure of the sample due to excessive pressure, the shear stress is applied step by step with an increment of 2% of the normal stress. For each shear stress level applied, the stable readings of the dial indicator of shear displacement and the digital pressure gauge of shear load are recorded, respectively, and then the next level of shear stress is applied until the sample is shear-damaged. The sign of shear failure of the sample is that the digital pressure gauge value of shear load no longer rises, and the shear displacement presents an accelerated or uniform state [38]. The relationship curves of shear

displacement and shear stress of root–soil composites with thick or fine roots corresponding to *RAR* = 0.1% under four normal stresses are shown in Figures 5 and 6, respectively.

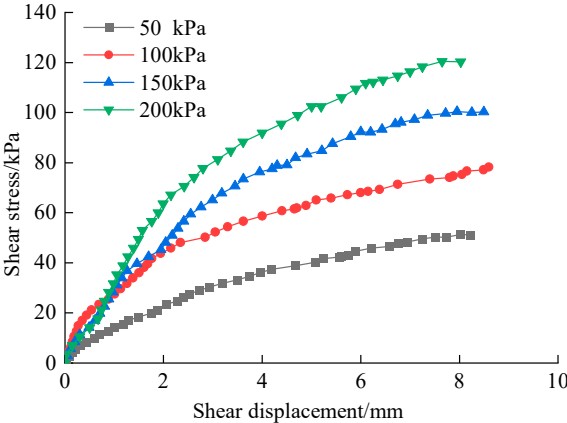

**Figure 5.** Relationship curves between shear displacement and shear stress of root–soil composite of fine roots with *RAR* = 0.1%.

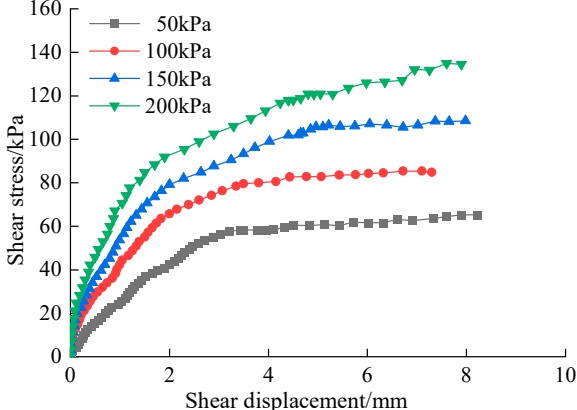

**Figure 6.** Relationship curves between shear displacement and shear stress of root–soil composite of thick roots with *RAR* = 0.1%.

## 4. Results and Analysis

### 4.1. Statistical Result of RAR

The statistical results of the distribution of the average diameter and number of *Cryptomeria japonica* (Thunb. ex L.f.) D.Don roots with the increase of buried depth under the two profiles are shown in Figure 7.

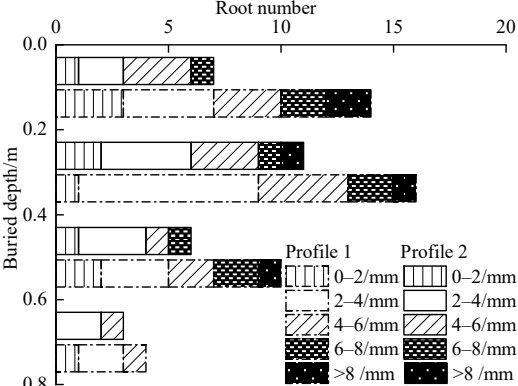

**Figure 7.** Distribution of root number and diameter with the increase of buried depth.

As seen from Figure 7, the number of roots in both profile 1 and 2 of *Cryptomeria japonica* (Thunb. ex L.f.) D.Don increases first and then decreases with the increase of buried depth. Among them, the number of roots in buried depth of 0.2–0.4 m is the highest, accounting for 36.36% and 40.74% of the total number of roots in profile 1 and 2, respectively. The number of roots in buried depth of 0.6~0.8 m is the fewest, accounting for only 9.09% and 11.11% of the total number of roots in profile 1 and 2, respectively. No root distribution is observed when the buried depth exceeds 0.8 m.

In addition, analysis of root diameters at different buried depths shows that root diameters of 2–4 mm and 4–6 mm are distributed in different depths of the two profiles, and root diameter larger than 6 mm is not distributed in buried depths greater than 0.6 m of both the two profiles. In the buried depth of 0.2–0.4 m, four root diameters are all distributed in both of the two profiles, and the number of root diameter of 2–4 mm is the highest, accounting for 50.00% and 36.36% of the total root number in profile 1 and 2, respectively.

Overall, the number of roots and the distribution range of root diameter in profile 2 are both smaller than that in profile 1, indicating that the root number and root diameter range of *Cryptomeria japonica* (Thunb. ex L.f.) D.Don decreases with the increase of distance from the trunk.

The *RAR* can be calculated using Equation (1) based on the number and diameter of roots measured at different buried depths. The relationship between the average *RAR* and buried depth for each plant in the two profiles is shown in Figure 8.

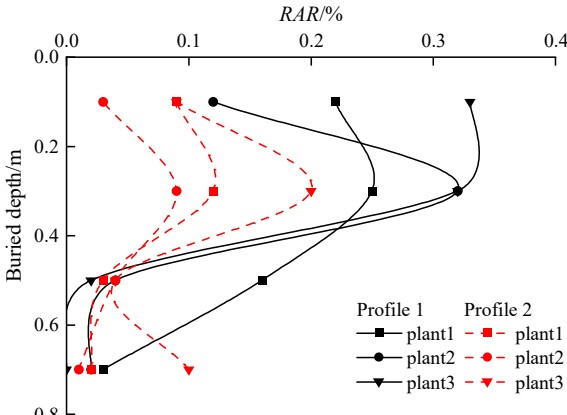

**Figure 8.** Distribution of *RAR* with the increase of buried depth.

It can be seen from Figure 8 that the distribution range of *RAR* is 0.0%~0.4%. In both profile 1 and profile 2, *RAR* increases first and then decreases with the increase of buried depth, and the maximum value of *RAR* appears in the buried depth of 0.2–0.4 m. The *RAR* values of different depths corresponding to profile 2 are generally lower than those corresponding to profile 1, indicating that the *RAR* value decreases significantly with the increase of distance from the trunk, especially in the buried depth of 0–0.4 m. At the same time, it can be found that the *RAR* values of different samples of *Cryptomeria japonica* (Thunb. ex L.f.) D.Don in the same buried depth are significantly different. For example, the *RAR* values corresponding to profile 1 of three plants in the buried depth of 0–0.2 m are 0.12%, 0.22%, and 0.33%, respectively. This indicates that even under similar topographic and geological conditions, there are still some differences in root distribution of *Cryptomeria japonica* (Thunb. ex L.f.) D.Don.

### 4.2. Analysis of Shear Strength Results of Root–Soil Composite

4.2.1. Shear Strength of Root–Soil Composite

The large-box direct shear tests were conducted to the root–soil composites with thick and fine roots with five levels of *RAR* gradient under four levels of normal stress of 50, 100, 150, and 200 kPa, respectively, and the results of shear strength are shown in Table 4. As seen from Table 4, compared with rootless soil, the overall growth rate in the shear strength of root–soil composite of *Cryptomeria japonica* (Thunb. ex L.f.) D.Don is 0.00%~41.30%, in which the growth rate of root–soil composite with thick roots is 5.43%~41.30%, and that of root–soil composite with fine roots is 0.00%~23.91%. The maximum growth rate of shear strength of root–soil composite with thick roots is 1.73 times that of root–soil composite with fine roots. The results show that both thick and fine roots can improve the soil shear strength, and the thick roots show a better strengthening effect than the fine roots.

**Table 4.** Shear strength and its increment of root–soil composites under different *RAR*s.

| RAR | Shear Strength and Its Increment/kPa | | | | | | | | | | | | | | | |
| | σ = 50 kPa | | | | σ = 100 kPa | | | | σ = 150 kPa | | | | σ = 200 kPa | | | |
| | Fine Root | Increment | Thick Root | Increment | Fine Root | Increment | Thick Root | Increment | Fine Root | Increment | Thick Root | Increment | Fine Root | Increment | Thick Root | Increment |
|---|---|---|---|---|---|---|---|---|---|---|---|---|---|---|---|---|
| 0.1% | 51.00 | 10.87% | 65.00 | 41.30% | 78.00 | 6.85% | 84.00 | 15.07% | 100.00 | 8.70% | 108.00 | 17.39% | 120.00 | 9.09% | 135.00 | 22.73% |
| 0.2% | 57.00 | 23.91% | 59.00 | 28.26% | 80.00 | 9.59% | 82.00 | 12.33% | 103.00 | 11.96% | 107.00 | 16.30% | 125.00 | 13.64% | 132.00 | 20.00% |
| 0.3% | 48.00 | 4.35% | 54.00 | 17.39% | 76.00 | 4.11% | 82.00 | 12.33% | 98.00 | 6.52% | 100.00 | 8.70% | 115.00 | 4.55% | 128.00 | 16.36% |
| 0.4% | 47.00 | 2.17% | 53.00 | 15.22% | 75.00 | 2.74% | 80.00 | 9.59% | 93.00 | 1.09% | 97.00 | 5.43% | 110.00 | 0.00% | 120.00 | 9.09% |

Note: The shear strengths of the rootless soil samples under 4 levels of normal stress are 46, 73, 92, and 110 kPa, respectively. The increment is the result obtained by comparing the shear strength of root–soil composite with that of rootless soil.

4.2.2. Effect of Root Content on the Shear Strength of Root–Soil Composite

The relationship curves between root content and shear strength of root–soil composite are shown in Figure 9. As seen from Figure 9, with the increase of *RAR*, the shear strength of root–soil composites with thick and fine roots both increases first and then decreases under all four levels of normal stress. The shear strength of root–soil composite with thick roots under four levels of normal stress all appear maximum value when *RAR* = 0.1%, and the corresponding peak shear strengths were 65, 84, 108, and 135 kPa, respectively. The shear strength of root–soil composite with fine roots under four levels of normal stress all appear maximum value when *RAR* = 0.2%, and the corresponding peak shear strengths were 57, 80, 103, and 125 kPa, respectively. This indicates that *RAR* significantly affects the shear strength of root–soil composite. The shear strength of root–soil composite shows a nonlinear relationship with the increase of *RAR*, and there exists an optimal *RAR* for the shear strength of root–soil composite to reach the maximum value—that is, there exists an optimal root content corresponding to the best reinforcement effect on soil. The optimal root content of thick roots is smaller than that of fine roots for *Cryptomeria japonica* (Thunb. ex L.f.) D.Don, which shows that the optimal root content decreases with the increase of root diameter.

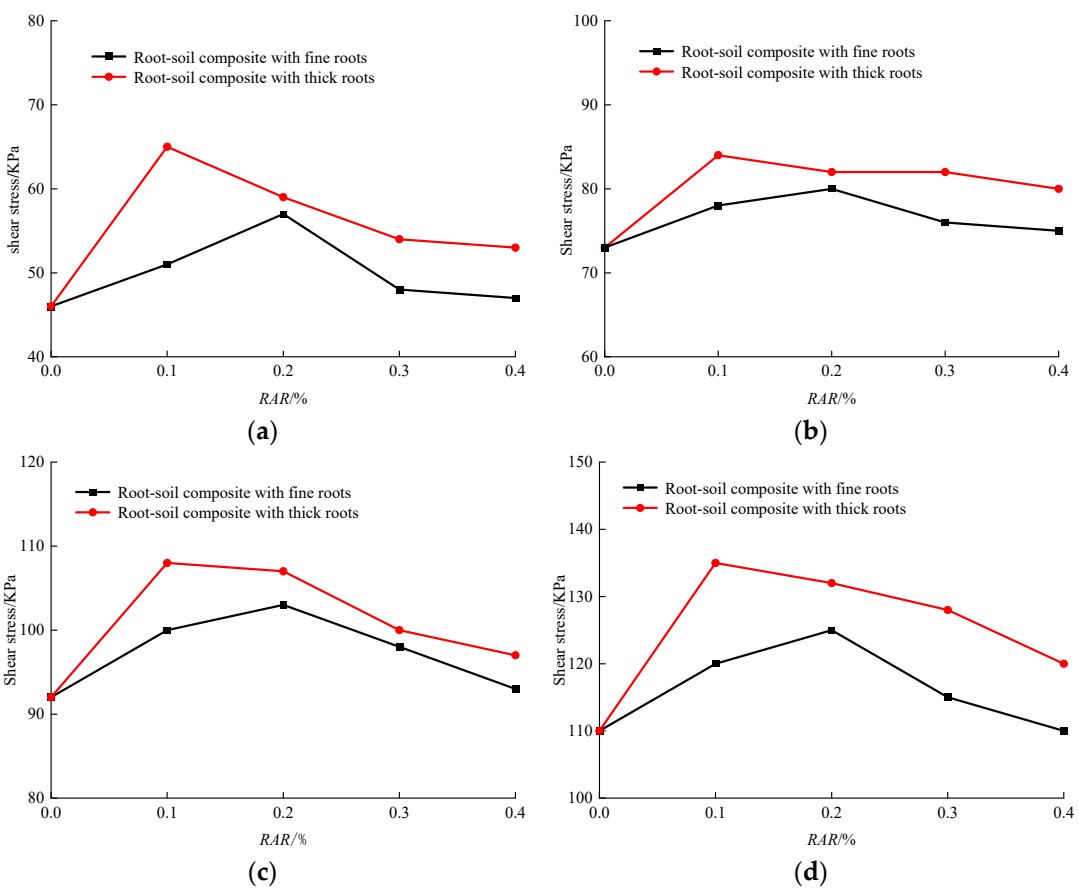

**Figure 9.** Relationship curves between *RAR* and shear strength of root–soil composite: (**a**) Normal stress of 50 kPa; (**b**) Normal stress of 100 kPa; (**c**) Normal stress of 150 kPa; and (**d**) Normal stress of 200 kPa.

### 4.2.3. Effect of Root Content on the Shear Strength Index of Root–Soil Composite

It can be seen from Table 4 that the shear strength of root–soil composite increases with the increase of normal stress. The shear strength of root–soil composite with thick and fine roots and normal stress can be linearly fitted under different *RAR* conditions, and the results are shown in Table 5. It can be found that the determination coefficients ($R^2$) are all greater than 0.99, indicating that the correlation is significant, and the form conforms to the Mohr–Coulomb strength criterion, as shown in Equation (2):

$$\tau = \sigma \tan \varphi + c \tag{2}$$

where $\tau$ is the shear strength (kPa); $\sigma$ is the normal stress (kPa); $c$ is the cohesion (kPa); and $\varphi$ is the internal friction angle (°).

Based on the Mohr–Coulomb strength criterion and fitting equations, the shear strength indexes (cohesion $c$ and internal friction angle $\varphi$) of root–soil composite with thick and fine roots under different *RAR* conditions can be obtained. The results are shown in Table 5.

As seen from Table 5, compared with rootless soil, the cohesion of root–soil composite for thick and fine roots increases by 3.50~12.00 kPa and 1.00~7.00 kPa, respectively, with the growth rate of 12.73%~43.64% and 3.64%~25.45%. The internal friction angle of root–soil composite for thick and fine roots increased by 0.70~3.10° and 1.10~1.70°, respectively, with the growth rate of 3.06%~13.54% and 4.80%~7.42%, except that of *RAR* = 0.4% is slightly lower than that of rootless soil. It can be found that the enhancement effect of roots on cohesion is greater than that on internal friction angle for soil by comparing the increasing range of cohesion and internal friction angle. It can be seen that the cohesion

has a significant effect on the shear strength of root–soil composite [39]. In addition, it can be found that the enhancement effect of thick roots on cohesion and internal friction angle is higher than that of fine roots.

**Table 5.** Shear strength indexes.

| RAR | Thick Root | | | Fine Root | | |
|---|---|---|---|---|---|---|
| | Fitted Equation | $c$/kPa | $\varphi$/° | Fitted Equation | $c$/kPa | $\varphi$/° |
| 0.1% | $\tau = 0.468\sigma + 39.5(R^2 = 0.994)$ | 39.5 | 25.1 | $\tau = 0.458\sigma + 30.0(R^2 = 0.995)$ | 30.0 | 24.6 |
| 0.2% | $\tau = 0.488\sigma + 34.0(R^2 = 0.999)$ | 34.0 | 26.0 | $\tau = 0.454\sigma + 34.5(R^2 = 0.999)$ | 34.5 | 24.4 |
| 0.3% | $\tau = 0.480\sigma + 31.0(R^2 = 0.993)$ | 31.0 | 25.6 | $\tau = 0.446\sigma + 28.5(R^2 = 0.988)$ | 28.5 | 24.0 |
| 0.4% | $\tau = 0.436\sigma + 33.0(R^2 = 0.993)$ | 33.0 | 23.6 | $\tau = 0.414\sigma + 29.5(R^2 = 0.984)$ | 29.5 | 22.5 |

Note: The cohesion and internal friction angle of the rootless soil samples are 27.5 kPa and 22.9°, respectively.

Figure 10 shows the relationship curves between *RAR* and shear strength indexes (cohesive *c* and internal friction angle *φ*) of root–soil composite. As shown in Figure 10a, it can be found that the cohesion of root–soil composite with thick and fine roots increases first and then decreases with the increase of *RAR*, and the maximum value appears when *RAR* = 0.1% and *RAR* = 0.2%, respectively. This trend is consistent with that of shear strength of root–soil composite and *RAR*. This consistency is mainly because the roots provide the root–soil composite with "seemingly cohesive" through root–soil interactions. The increase in the shear strength of root–soil composite is mainly due to the increase in the cohesion of root–soil composite, which is similar to the research results of Hu et al. [28] and Chen et al. [40].

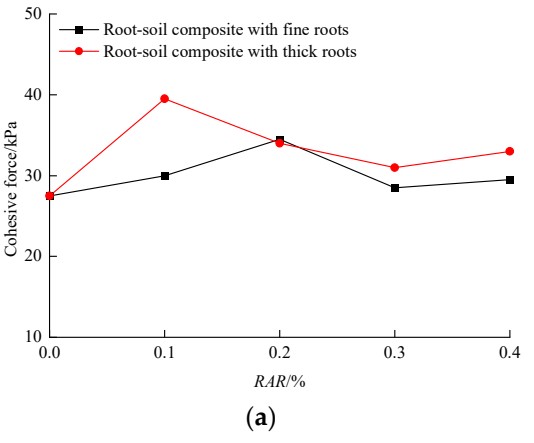

(**a**)

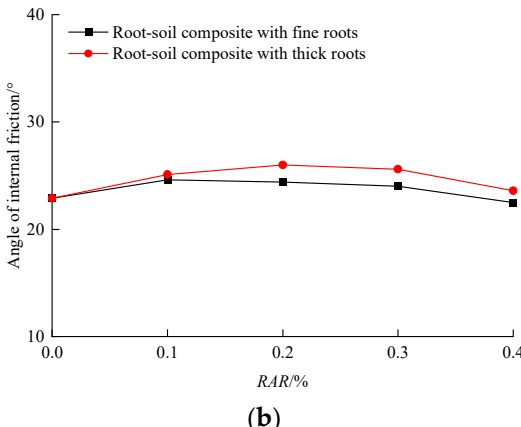

(**b**)

**Figure 10.** Relationship curves between *RAR* and shear strength indexes of root–soil composite: (**a**) *RAR* and cohesive and (**b**) *RAR* and internal friction angle.

It can be seen from Figure 10b that with the increase of *RAR*, the internal friction angle of root–soil composite with thick and fine roots both increases first and then decreases, but the change trend is relatively gentle and the difference is not significant.

## 5. Analysis of RAR Influence Mechanism

Shear strength of root–soil composite is one of the important indexes to evaluate the mechanical effect of soil consolidation and slope protection with root system, and root content is one of the important factors affecting the shear strength of root–soil composite [41]. In this study, there exists an optimal *RAR* to maximize the shear strength of root–soil composite, i.e., the existence of an optimal root content makes the soil reinforcement effect to achieve optimal result.

When the root content is less than the optimal root content, the shear strength of root–soil composites with thick and fine roots both increases with the increase of root

content, and the shear strength of root–soil composites with thick roots is greater than that of root–soil composites with fine roots. In order to investigate the difference between thick and fine roots in improving the soil shear strength, the shear failure sample was carefully stripped to obtain the map of damaged root at the shear plane, as shown in Figure 11. It can be found that the thick roots have shear damage at the shear plane, while the fine roots have no obvious damage at the shear plane, and the roots are not pulled off in root–soil composite, indicating that the roots of *Cryptomeria japonica* (Thunb. ex L.f.) D.Don mainly enhance the shear strength of the root–soil composite through the friction effect at the root–soil interface.

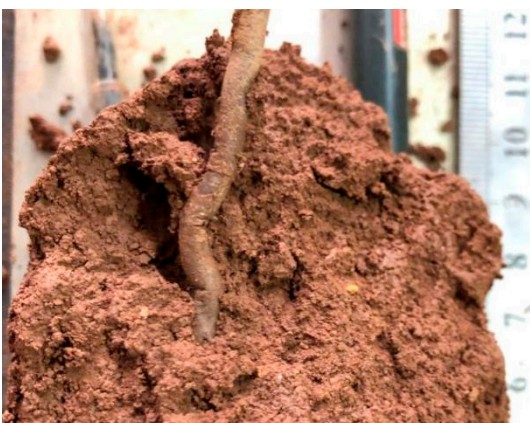

**Figure 11.** Map of damaged root.

When the root content exceeds the optimal root content, the shear strength of root–soil composites with thick and fine roots both decreases with the increase of root content, and the former is greater than the latter. The reason is that under the action of shear load, the soil on the shear plane is subjected to shear stress, and soil particles gradually develop into directional arrangement from close packing. The soil structure is destroyed, and the displacement of the soil along the shear direction gradually increases, thus accumulating to form a through-slip surface, resulting in shear failure of the soil. With the increase of *RAR*, the roots in the sample increases. When the root–soil composite is subjected to shear load, the roots and soil cause uncoordinated deformation due to the large difference in their shear capacity, which makes the soil particles around the root–soil contact surface easier to rotation, sliding, and rearrangement, accelerating the structural failure and through-slip of the soil on the shear plane, and thus reduces the shear strength of the root–soil composite.

In addition, it can be found that under the same root content conditions, the thick roots have better shear capacity than the fine roots, and can better resist the shear stress of the sample, which results in the shear strength of root–soil composite with thick roots is better than that of fine roots.

The optimal root contents corresponding to the root–soil composites with thick and fine roots are 0.1% and 0.2%, respectively, and the former is smaller than the latter. As mentioned in the introduction, when the soil is reinforced with the roots of herbs and shrubs, the corresponding optimal root content varies in the range of 3% to 5% [32], whereas Liu et al. [38,42] conducted direct shear tests of root–soil composites on the roots of *Bischofia javanica* Blume and *Malus halliana* Koehne and found that the optimal root content was 0.2%. It can be found that the optimal root content corresponding to herbs and shrubs is greater than that of arbors, and the optimal root content of *Cryptomeria japonica* (Thunb. ex L.f.) D.Don in this study is closer to that of arbors such as *Bischofia javanica* Blume and *Malus halliana* Koehne. At the same time, it can be found that the optimal root content tends to decrease with the increase of root diameter. The reason may be that the existence of roots will affect the soil in a certain range around the roots. With the increase of root diameter, the influence range also increases. Compared with fine roots, thick roots have stronger shear capacity, larger root diameter, and wider influence range. With more roots,

the influence range of roots on the surrounding soil overlaps more quickly, thus achieving the optimal root content under smaller *RAR* conditions. Limited to experimental means, further validation is needed in the later stage.

## 6. Discussion

In this study, the effect of root content on the shear strength of root–soil composite with thick and fine roots of *Cryptomeria japonica* (Thunb. ex L.f.) D.Don is briefly discussed based on the improved direct shear test, but limited to the complexity of the research problem, many simplifications have been made in the test process, resulting in some limitations, specifically as follows:

(1) Remolded soil was used for sample preparation in this study. Remodeled soil destroys the original soil structure and changes the original cross-binding state between the roots and soil, which will have a great impact on the test results. It can be supplemented and verified by carrying out in situ tests or collecting original root–soil composite specimens for laboratory testing in the later stage.

(2) The shear surface area of the direct shear instrument used in this study is small, only 15 cm × 15 cm, and the test results may be affected by the size effect. In order to eliminate the influence of the above factor, a large direct shear instrument is considered for subsequent tests.

(3) Compared with engineering measures, the reinforcement effect of plant measures on slope is mainly limited to shallow surfaces. In addition to considering the influence of root systems on soil, it is also necessary to consider the influence of plants on slope hydrology. Therefore, when evaluating the slope protection effect of plants, it is necessary to conduct comprehensive research combined with rainfall infiltration test.

## 7. Conclusions

(1) The *RAR* of *Cryptomeria japonica* (Thunb. ex L.f.) D.Don increases first and then decreases with the increase of buried depth, and the maximum value appears in buried depths of 0.2–0.4 m, while the *RAR* decreases significantly with the increase of horizontal distance from the excavation point to the trunk.

(2) Both the thick and fine roots of *Cryptomeria japonica* (Thunb. ex L.f.) D.Don can increase the soil shear strength under all four levels of normal stress; the thick roots are superior to the fine roots, and the maximum growth rate of the former is 1.73 times that of the latter.

(3) With the increase of *RAR*, the shear strengths of root–soil composites with thick and fine roots of *Cryptomeria japonica* (Thunb. ex L.f.) D.Don both increase first and then decrease under all four levels of normal stress, which means that there exists an optimal root content for the roots with the best reinforcement effect on soil, and the optimal *RAR* corresponding of the two is 0.1% and 0.2%, respectively.

(4) The cohesive and internal friction angle of the root–soil composite can be improved by the roots of *Cryptomeria japonica* (Thunb. ex L.f.) D.Don, and the thick root is better than the fine root. Meanwhile, the enhancement effect on the cohesion is greater than that on the internal friction angle. With the increase of *RAR*, the cohesion of root–soil composites with thick and fine roots both increases first and then decreases, and the change trend is consistent with the change trend of *RAR* and shear strength of root–soil composite, but the corresponding internal friction angle of the two does not differ much.

**Author Contributions:** Conceptualization, J.L.; methodology, Y.T.; software, Y.T.; investigation, Y.T., Y.J. and S.L.; data curation, Y.T.; writing—original draft, Y.T., Y.J. and S.L.; writing—review and editing, J.L., X.P., Y.P. and K.W. All authors have read and agreed to the published version of the manuscript.

**Funding:** This research was funded by the National Natural Science Foundation of China, grant Nos. 41907242 and 42177140.

**Data Availability Statement:** Data are contained within the article.

**Conflicts of Interest:** Kai Wu was employed by Sichuan Highway Planning, Survey, Design and Research Institute Ltd. Xingxin Peng was employed by China Railway 11th Bureau Group Fourth Engineering Co., Ltd. The remaining authors declare that the research was conducted in the absence of any commercial or financial relationships that could be construed as a potential conflict of interest.

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
