# Peer review of "Experimental Study on the Effect of Root Content on the Shear Strength of Root–Soil Composite with Thick and Fine Roots of Cryptomeria japonica (Thunb. ex L.f.) D.Don"

_forests, doi:10.3390/f15081306_

Round 1

Reviewer 1 Report

Comments and Suggestions for Authors

Several questions and remarks regarding the paper by Liu et al. “Experimental study on the effect of root content on the shear strength of root-soil composite with thick and fine roots of Cryptomeria fortunei”.

Please include information on the slope grade because the research was conducted on Boshan Mountain, which has a slope.

With depth, the physical properties of the soil change considerably. Table 1 presents data for which soil layer? Describe the physical properties of the various soil layers.

“The overall excavation depth is about 1 m.” (Line 152). Data are displayed in Figure 7 up to 0.8 metres. Specify how deep the samples were taken.

“Since the tree age could not be accurately estimated…” (Line 141). Why? The precise age of the tree can be determined with the aid of increment borer chosen cores and a straightforward ring counting method. What prevented you from using dendrochronological techniques to establish the precise age of the trees?

The authors must correctly specify the Latin names of the plants. First, you must provide an accepted biological name, not synonyms: Cryptomeria fortunei - this name is a synonym of Cryptomeria japonica (Thunb. ex L.f.) D.Don (accepted Latin name), Inula viscosa - this name is a synonym of Dittrichia viscosa (L.) Greuter (accepted Latin name), Nitraria tangutorum- this name is a synonym of Nitraria sibirica (DC.) Pall. (accepted Latin name). Second, the first mention of the species' Latin name in the text should be given in full.

Paper mulberry. (Line 63). This is not the Latin name of the species. The accepted Latin name for this plant is Broussonetia papyrifera (L.) L'Hér. ex Vent.

Add research limitations to the Discussion section.

Reviewer 2 Report

Comments and Suggestions for Authors

This is a unique paper about the strength of the root systems of Chinese cedar, but the following perspective is missing, so please add additional explanation.

Based on the growth characteristics of Chinese cedar, there are soils that are suitable for its growth. What soil type was considered in this study?

In addition, soil has elements such as soil texture, soil structure, and soil hardness. Please add a note about these points as well.

Reviewer 3 Report

Comments and Suggestions for Authors

This paper is about Experimental study on the effect of root content on the shear 2 strength of root-soil composite with thick and fine roots of 3 Cryptomeria fortunei. This paper mentions that:

In order to investigate the difference in shear strength of root-soil composite with thick and fine roots under different root content conditions, Cryptomeria fortunei was selected as the research object, and the distribution characteristics of its roots with the increase of buried depth was studied using longitudinal profile method. Based on the distribution range of root area ratio (RAR) in field investigation, the modified large-scale direct shear tests were carried out on the root-soil composite samples with thick and fine roots of 5 RAR grades under 4 normal stress levels, and the variation rule of the shear strength of the root-soil composites with thick and fine roots under different root contents was analyzed. The influence mechanism of RAR was briefly discussed. The results show that the RAR of Cryptomeria fortunei increases first and then decreases with the in-crease of buried depth, and decreases with the increase of horizontal distance from the excava-tion point to the trunk. Both the thick and fine roots can increase the shear strength of soil, but the effect of thick roots is greater than that of fine roots. The shear strengths of root-soil composites with thick and fine roots both increase first and then decrease with the increase of RAR, which means that there exists the optimal root content for the roots with the best reinforcement effect on soil, and the optimal RAR is 0.1% and 0.2%, respectively. The cohesive and internal friction angle of the root-soil composite can be improved by the roots of Cryptomeria fortunei, and the thick root is better than the fine root. Meanwhile, the enhancement effect on the cohesion is greater than that on the internal friction angle. The results are of great significance for understanding the effect of roots on shear strength of soil and enriching the existing theory of slope protection with arbors.

In the opinion of the reviewer, this manuscript note could be accepted after the major revisions and reevaluation.

1-     Mention more about novelty of this research.

2-     The abstract section is proposed to be reviewed and rewrite. Mention more details in the abstract and delete additional information.

3-     It is recommended to use high quality figure (Figure 1)

4-     It is recommended to prepare a table in the introduction section and mention list the literature works done in this regard in order of year.

5-     Some of the references provided are old. It is suggested that a number of articles about related topics.

6-     The conclusion section is suggested to be rewritten. It is suggested to remove numbering in the conclusion section and use bulleting.

7-     According to iThenticate report, percent of match is 34%. It seems that in some parts it needs to be rewritten and reduce similarity index.

Round 2

Reviewer 1 Report

Comments and Suggestions for Authors

All the responses and revisions made by the authors are clear, thoughtful, and satisfactory. Thanks to the authors.

Reviewer 3 Report

Comments and Suggestions for Authors

Accepted.